# The Implications of Family Members’ Absence from Hospital Visits during the COVID-19 Pandemic: Nurses’ Perceptions

**DOI:** 10.3390/ijerph19158991

**Published:** 2022-07-24

**Authors:** Tânia Sofia Pereira Correia, Maria Manuela F. P. S. Martins, Fernando F. Barroso, Lara G. Pinho, César Fonseca, Olga Valentim, Manuel Lopes

**Affiliations:** 1Instituto de Ciências Biomédicas Abel Salazar (ICBAS), Universidade do Porto (UP), 4050-313 Porto, Portugal; 2CINTESIS (Centro de Investigação em Tecnologias e Serviços de Saúde)—NursID (Innovation & Development in Nursing), 4050-313 Porto, Portugal; mmartins@esenf.pt (M.M.F.P.S.M.); olga.valentim@ipluso.pt (O.V.); 3Escola Superior de Saúde Ribeiro Sanches (ERISA), Instituto Politécnico da Lusofonia(IPLUSO), 1950-396 Lisboa, Portugal; 4Nursing Department, Universidade de Évora, 7000-811 Évora, Portugal; lmgp@uevora.pt (L.G.P.); cfonseca@uevora.pt (C.F.); mjl@uevora.pt (M.L.); 5Escola Superior de Enfermagem do Porto (ESEP), 4050-313 Porto, Portugal; 6Centro Hospitalar de Setúbal, 2910-446 Setúbal, Portugal; faustobarroso@gmail.com; 7Comprehensive Health Research Centre (CHRC), Universidade de Évora, 7000-811 Évora, Portugal; 8Escola Superior de Saúde de Leiria (ESSLei), Instituto Politécnico de Leiria (IPLeiria), 2411-901 Leiria, Portugal

**Keywords:** family nursing, family-centered care, hospitalization, COVID-19, patient safety, safety management

## Abstract

Background: In response to the COVID-19 pandemic, several measures were taken to prevent the transmission of infection in the hospital environment, including the restriction of visits. Little is known about the consequences of these directives, but it is expected that they will have various implications. Thus, this study aimed to understand the consequences of measures to restrict visits to hospitalized individuals. Methods: A qualitative interpretive study was conducted through semistructured interviews with 10 nurses chosen by convenience. Content analysis was performed using Atlas.ti software, version 22 (Berlin, Germany). Results: Twenty-two categories and eight subcategories were identified and grouped according to their scope: implications for the patient, implications for the family, and implications for care practice. Conclusions: The identified categories of implications of restricting hospital visits (implications for patients, relatives, and care practices) are incomparably more negative than positive and have a strong potential to cause safety events in the short to long term, also jeopardizing the quality of care. There is the risk of stagnation and even setback due to this removal of families from the hospital environment, not only in terms of safety and quality of care but also with regard to person- and family-centered care.

## 1. Introduction

Since the emergence of the first case of coronavirus disease 2019 (COVID-19), its prevalence continues to increase on a global scale [1]. Its impact on the various domains of health care organizations is undeniable, being considered by the World Health Organization (WHO) [2] as the largest global public health emergency and as being responsible for extreme pressure on health systems [3].

To maintain safety in the process of health care by preventing the transmission of infection, particularly at the hospital level, several containment measures were taken, including the restriction of visits by relatives and other important visitors to hospitalized patients throughout the world [4]. The restriction of hospital visits is as old as the first hospitals and has emerged in association with the need to contain diseases and protect patients and communities [5]. The social changes during the 1960s associated with increased awareness of the advantages to the patient of being able to receive family visits led hospital administrations to review the visit policies, and since then, more open models have emerged [6].

Currently, there is some evidence that the presence of the family brings benefits to the hospitalized patient and to the care process [7,8,9]. There are several models and/or theories that define and substantiate levels of family involvement in the care of the hospitalized person, as is the case of the patient and family-centered care (PFCC). PFCC is an approach based on reciprocally beneficial partnerships between healthcare providers, patients, and families in the healthcare process [10]. This conception of care is based on the definition of patients and family members as essential allies for the quality and safety of health care. The available evidence has revealed that PFCC has better health outcomes, namely, the reduction in hospitalizations and errors in health care, the improvement in the care experience, and satisfaction of the patient and family, and contributes to more effective management of resources [10,11]. In this sense, the planning of person-centered health care should be individualized, dynamic, flexible, and participatory, seeking to involve the family, respond to the specific needs of the person, and improve patient satisfaction [12].

The effectiveness or real contribution of this restriction of hospital visits in preventing the spread of COVID-19 is not fully known; the current evidence on the subject is scarce and identifies several other possible lines of in-hospital transmission in addition to visits [13,14]. Namely, a study conducted at University College London Hospitals on 1160 inpatient beds in four hospitals revealed that transmission by COVID-19 in a hospital setting occurred in 55% of hospitalized patients and 14% of cross-infected patients without contact but in the same ward or through equipment or professionals, and only in 12% was the source not identified, but it is thought that these may have originated from asymptomatic health professionals and/or visiting family members [14]. Although the existing evidence on the role of the family in the transmission of COVID-19 and other infections in the hospital environment is limited, the known data indicate that families do not play a predominant role in this transmission in hospitals [15,16]. Nevertheless, little is known about the consequences of visit restriction policies, either from the point of view of containing viral transmission or from the effects on the care process from the perspective of patients, family, and/or nurses [17]. However, it is expected that these directives have had several consequences, which are necessary to know [18]. The present study aimed to understand the implications of restricting visits to hospitalized individuals. The criteria for applying restrictions on visits due to COVID-19 varied from institution to institution. There were also times when there were statutory enactments of this restriction that was blind and did not consider the consequences of these measures. This study may help to ensure that the measures to restrict visits in response to this or any other situation are appropriate and in the best interest of the patient and the quality and safety of care. In this way, the study aims to answer the following research questions:

What are the implications of the absence of family members in hospital visits during the COVID-19 pandemic identified by nurses?

How do nurses assess these implications for the patient, family, and care practice?

## 2. Materials and Methods

The restriction of hospital visits due to the COVID-19 pandemic is a relatively recent phenomenon of a complex and multifactorial nature, so we performed a qualitative, interpretative study with a thematic analysis [19]. This study used the intentional sampling method. The participants of this study were nurses working during the COVID-19 pandemic period in four hospitals in northern Portugal in inpatient internal medicine and surgical services who were selected by convenience because they were the most accessible and met the preestablished inclusion criteria [20]. Thus, the inclusion criteria were being a practicing nurse during the COVID-19 pandemic period; practicing in inpatient internal medicine and adult surgical services in one of the four hospitals in northern Portugal (in any specialty); having at least 4 years’ experience in these hospitals (this criterion arises from the authors’ understanding as the minimum experience to be able to contribute to the study considering that it is a topic that may not be the focus of attention of less experienced professionals.); and being available to participate.

After the seventh interview, no new data were found that are new categories, so after ten interviews, we considered that data saturation had been reached. Based on this criterion of theoretical saturation, we reached a total of 10 participants: 9 female nurses and 1 male nurse, aged between 28 and 62 years (mean of 39 years), 1 participant under the age of 30, 6 between the ages of 30 and 40, 2 participants between the ages of 40 and 50, and 1 participant over 60, with an experience of 6.5 to 39 years (mean of 16 years). Seven of the respondents had a master’s degree, and nine were specialists (six in rehabilitation nursing, two in medical-surgical nursing, and one in mental health nursing and psychiatry).

Data were collected through individual semistructured interviews between June and September 2020 to learn the perceptions of nurses about the implications of the absence of families from hospitals during the COVID-19 pandemic with the following questions:

What do you see as positive and negative in restricting visits?

What was favorable and unfavorable of the restriction of visits for nurses?

What was favorable and unfavorable of restricting visits for patients?

Each interview was transcribed in full, and its transcript was sent to the interviewee for validation (Figure 1). The content analysis was developed with thematic analysis [19], which defines three phases: preanalysis, exploration, and finally, the treatment of results and interpretation [19]. In the exploratory analysis, common contents were searched for, which allowed the identification of thematic areas designated by categories. Atlas.ti^®^software was used in the coding and categorization of the interviews. We used this software to create the units of analysis (categories) and identify them from the significant excerpts of the interviews that were grouped into large thematic areas (families), which enabled the obtain of the results of this study.

Ethical and legal principles were respected. The study was authorized by the Joint Ethics Committee of the Hospital and University Center of Porto and the Biomedical Sciences Institute Abel Salazar (Instituto Ciências Biomédicas Abel Salazar—ICBAS) of the University of Porto (Universidade do Porto—UP). All procedures performed with the participants respected anonymity, confidentiality, and informed consent, as well as the Declaration of Human Rights of Helsinki.

## 3. Results

From the analysis of the data obtained in the interviews of the nurses, 22 categories (codes) and 8 subcategories were identified, which were grouped into 3 major thematic areas (families) according to the scope of the implications: implications for the patient, implications for the family, and implications for care practice (Figure 2, Figure 3 and Figure 4, diagrams drawn with Atlas.ti^®®®^ software).

Eight categories and four subcategories were found that were considered consequences for the patient: *Depersonalization* (1.1), *Jeopardized mental health* (1.2) with two subcategories: Sadness/depressed mood (1.2.1), Anxiety (1.2.2), Isolation/loneliness (1.2.3), and confusion/agitation (1.2.4), *Feeling of insecurity by the patient* (1.3), *Resistance of the patient to the therapeutic regimen* (1.4), *Greater desire of the patient for discharge* (1.5), *Lack of an affectionate relationship and emotional support by the family* (1.6), *Lack of stimulation from the family* (1.7), and *Feeling of abandonment* (1.8) (Figure 2).

As implications on the family members caused by the policy of restricting visits to hospitalized patients, five categories were identified: *No access to patient information* (2.1), *Anxiety* of the family (2.2), *Greater need for information* (2.3), *Removal of the family from assuming the role of caregiver* (2.4), and *Feeling of insecurity* (2.5) (Figure 3).

Nine categories were identified as implications for care practice, with four subcategories. Five categories were considered positive implications for care practice: *More time for nurses to care for the patient* (3,1), *More effective infection control* (3.2), *More peaceful infirmary* (3.3), *Nurses feeling safer from COVID-19* (3.4), and *Simplification/facilitation of care and delivery* (3.5). Four categories, with four subcategories, had negative implications for care practice: *Absence of family-centered care* (3.6) (subcategories: *Noninvolvement of the family in the care* (3.6.1) and *Difficulty training caregivers* (3.6.2)), *Communication with the affected family* (3.7) (subcategories: *Lack of information on patients* (3.7.1) and *Difficulty planning discharge* (3.7.2)), *Less time for providing care* (3.8), and *Less person-centered care* (3.9) (see Figure 4).

## 4. Discussion

### 4.1. Implications for the Patient

A consequence for the patient of the absence of families in the hospitals identified by the nurses was *Depersonalization* (1.1): “And we miss visits so much. When patients come from other hospitals, the relatives don’t come with the purpose of bringing pajamas or footwear, and then we have patients with disposable slippers with hospital clothes completely depersonalized. It’s sad” (nurse 1 (N1)). The depersonalization of the hospitalized patient is understood as the experience of identity loss resulting from the vulnerability of the hospitalized patient when separated from family and their usual context [21]. In this case, the separation from the family, in addition to not allowing the patient to feel him/herself with the help of his/her belongings, the very absence of these contributes to their feelings. The depersonalization of the patient as an individual with a history, family, and identity is described in the literature as harmful, distressing, and dehumanizing [22]. The burnout of health professionals associated with the COVID-19 pandemic can be a potentiator of this dehumanization process in the context of hospital health care [22].

The *Jeopardizing of the patient’s mental health* (1.2) is another implication of the absence of families from their hospitalized relatives, identified in the speeches of the participants: “…it’s very clear that the patients miss them a lot. Even for their mental health it’s very evident…” (N9). In this context, it was possible to identify the following subcategories: *Sadness/depressed mood (1.2.1)*, *Anxiety* (1.2.2), *Isolation/loneliness* (1.2.3), and Confusion/agitation (1.2.4). With regard to depressed mood and/or sadness, it was stated that “…in terms of the emotional aspect, they become sadder, more crestfallen, more anxious, especially the oriented patients…” (N8). Depression/sadness has repercussions on care practice: “This makes it harder to provide care because the patients are more depressed and sad” (N3). Regarding the anxiety caused by the absence of families, the nurses reported, “As a negative, I see increased anxiety in the patients themselves” (N5). Regarding this relationship, a study conducted on 70 hospitalized patients concluded that the presence of the family during invasive nursing procedures reduced patient anxiety and that nurses should optimize their presence as a nonpharmacological strategy for controlling anxiety [23]. Naturally, patients without visitors, including family members, feel alone, so several nurses mentioned isolation/loneliness as an important implication for the patient: “…the absence of families is negative because it has a very large impact on patients, because it creates nostalgia, a feeling of loneliness…” (N3). Confusion/agitation was also referred to as a consequence: “Patients get slightly more agitated because they don’t receive visits, and if they are confused patients, they don’t understand why they don’t receive visits…” (N8). There is some evidence that corroborates these findings and confirms that this containment strategy of COVID-19 leads to consequences on the mental health of hospitalized people, leading not only to an increase in depressive symptoms but also to loneliness, agitation or even aggressiveness, cognitive worsening, and general dissatisfaction [17]. The last three consequences were not mentioned by our participants, although dissatisfaction was implicit in many identified categories. This fact may show that patient satisfaction is not valued by the nurses interviewed, as are the other explicit categories. Hospitalization alone generates stress and anxiety and can have several repercussions in terms of mental health, given that the person is in a vulnerable state with fear for their health. If we add the fact that we cannot have the in-person support of family members and friends, it would be expected that the restriction of visits would have repercussions on mental health. Psychosocial support by the family during hospitalization is important to the patient [24].

Another category identified was the *Feeling of insecurity by the patient* (1.3). The nurses interviewed reported that “The removal of families from the hospital continues to be disadvantageous, especially for the patients. There is a lot of insecurity for both; that is, the patients feel more alone and therefore more insecure…” (N10). This feeling of insecurity by patients is corroborated by studies that indicate that isolating patients in their most vulnerable moments from the people who know them best places them at risk of medical error and inconsistent, unnecessary, and expensive health care interventions [25].

*Resistance of the patient to the therapeutic regimen* (1.4) in the absence of the family in the hospital context was also identified: “it led… to resistance to treatment adherence. The support of the family is necessary, and its absence leads to patients not having an attitude of adherence” (N2). Similarly, several studies conclude that family support is an important factor in adherence to the therapeutic regimen [26,27,28].

In turn, the discomfort associated with the separation from their families is described by the interviewees as causing *Greater desire of the patient for discharge* (1.5): “…especially oriented patients, more autonomous, who can even contact, on their own, their family members by cell phone, who at this point become more anxious and voice greater appeals for discharge” (N8). This fact can affect discharge planning and contribute to adverse events associated, for example, with medication at home [29]. Thus, it can affect management and adherence to the therapeutic regimen after discharge.

The *Lack of an affectionate relationship and emotional support by the family* (1.6) was reported as important for the patient: “Family support is necessary for the patient…. Emotional support by the health team and emotional support by the family are different” (N2). In addition to the *Lack of stimulation from the family* (1.7), “there are many limitations, especially for these patients with decreased consciousness and who need stimulation from the family. I think it’s a crime not to have the family with the patient…. We have a patient who has reduced consciousness, is in a coma, and for example, in this context it is practically impossible to do sensory stimulation, we do not know anything about the patient, his habits. We can’t even give him a known voice, we can make a video call, but little else” (N1). That is, at the cognitive level, there is evidence that the separation of families from patients can lead to a worsening of cognitive functioning [17].

The *Feeling of abandonment* (1.8) was often reported by the interviewees: “The patients who are elderly, some more confused, do not understand why they are never visited. Some even say that ‘they abandoned me here, they left me here, they don’t care about me.’ Although we have and have made video calls and still continue to make video calls with some frequency, they still don’t realize it, so this is a negative aspect of the absence of visits” (N8).

It can be seen that the nurses interviewed were unanimous in stating that for the patient, the implications of the COVID-19 visit restriction policy were negative for hospitalized patients in several domains, including their mental health and adherence to the prescribed therapeutic regimen.

### 4.2. Implications for the Family

The nurses participating in this study report that families lose access to the necessary information about their hospitalized relatives (2.1): “Another feedback that families give us is that they can’t get much information about the patients, they can’t talk to us on the phone, can’t speak with the doctors” (N4), demonstrating the category *No access to patient information* (2.1).

This separation of the families from their hospitalized family members, as well as the lack of information mentioned above, leads to family anxiety (2.2). The participants reported that “As a negative, I see increased anxiety for the patient, and insecurity not only for the patient but also for the family, who end up not having information about the patient’s situation” (N5). The consequent anxiety about these restrictive measures by the family is described in several studies as having an impact on their functioning and well-being [17].

This anxiety and uncertainty about their hospitalized family member are clearly identified as the cause of the category *Greater need for information by families* (2.3) [17]. The participants mentioned this need as having an impact on the work of the hospital: “Because people couldn’t visit, they called many times, and so practically, a lot of the time we are answering the phone to provide information, which is legitimate, understandable. We had several family members of the same patient calling, and sometimes 5 min apart. There is no connection between them outside the hospital, but they end up overloading the service in this sense as well” (N8).

Finally, the *Removal of the family from assuming the role of caregiver* (2.4) is an implication of these measures with long-term repercussions: “Basically, this even interferes with the role of wanting to be a caregiver because people can’t understand the responsibilities they are taking, or can’t facilitate it because they have no idea, or don’t want to because they’re frightened by what they don’t know, and we also have this role of demystifying the situation of the patient a little and showing how they are” (N1). Evidence on this subject is still scarce, and further studies are needed to understand the long-term implications of the restrictive measures.

The *Feeling of insecurity* (2.5) by the family was also identified: “The removal of families from the hospital continues to be disadvantageous, especially for the patients; there is a lot of insecurity for both, that is, the patients feel more alone and therefore more insecure, and families, it is unimaginable what many families are going through because they can’t see their relatives, no matter how many alternatives may exist” (N10). This is an important category because these restrictive measures are enforced on the basis of the safety argument, yet families feel less secure. The existing evidence on the role of the family in the transmission of COVID-19 in the hospital setting is limited. A systematic review based on cases from Wuhan, China, before the implementation of restrictive measures showed that only 2% of the spread of the virus in the hospital environment was due to external people, such as visitors [15]. On the other hand, visit restriction policies are associated with multiple risks for patients and families [16,24,30,31,32]. Family members are essential, particularly in situations where patients are disabled, as they act as patient advocates and contribute to their safety [33].

Thus, it appears that the measures implemented to restrict visits, according to the reports of the participating nurses, proved to have negative impacts on the families, lowering their well-being, increasing their anxiety, and increasing their appeal for information that they cannot obtain easily, so they feel insecure about the care provided to their hospitalized relative.

### 4.3. Implications for Care Practice

The nurses reported in their interviews that the absence of families brought more positive time for nurses to care for the patient (3.1): “In a way, this decreases the request of the family members to nurses for information, culminating in more time available for nurses to provide care to the patients, and I consider this to be a positive…” (N5). Regarding this category, it is not clear what motivation leads the nurses to identify the family as a time consumer and not an investment of their time as part of the practice of caring for the patient and family. The high workload evidenced by the low ratios here compared to in other European countries may suggest a care practice under greater time pressure; therefore, nurses prioritize more basic care over other roles such as family-centered care [34].

As widely stated, these measures were taken to contain the transmission of COVID-19, so it is natural that the nurses interviewed consider that with the removal of visits, *More effective infection control* (3.2) can be achieved: “…we have many fewer people in the service, our service is small, a very old ward, with little space between beds. There is a problem with infection control” (N4). Nevertheless, as mentioned above, the existing evidence on the role of the family in the transmission of COVID-19 and other infections in the hospital setting is limited, but the known data indicate that family visitors do not play a great role in its transmission [15,16].

Another positive implication on care practice identified was the quieter infirmary (3.3), where the nurses stated that “As a positive, it was basically good for the nurses, and for assistants. It helps us a lot in our dynamics since we have those usual routines of an internment of medicine and ends up facilitating our work a little. Not having to arrive at that time and having to be constantly asking the visitors to give us a space to provide hygienic care, the positioning of the patients…. The service is much quieter, sometimes it is impressive during the afternoon of a weekend, some people find it easier to visit relatives at that time, and in a ward with eight patients, following the rules, each patient has the right to two visitors, that is, 16 people in that room, plus the patients, plus us. It is a lot of people, and at some point it is very confusing even for the patients, even for us it is a lot of confusion. And so I confess that my colleagues and I were very calm with the restriction of visits” (N8). This testimony shows that nurses found that hospital structural conditions are not ideal for accommodating the stipulated number of visits, which determines how they see the patient’s family and their care practice itself. Thus, essential questions about who should be present and who should participate in the care practice are not valued or discussed. It also shows that the policies of visits before COVID-19 were poorly structured for ensuring the safety and quality of care, so it will be important to rethink them regarding the aforementioned dimensions. It is important that the structural and procedural conditions allow the involvement of the family in care so that professionals do not see the family as a hindrance to the provision of care but as a coproducer of care.

The nurses also felt more secure about COVID-19 (3.4) with the restrictions: “…what we feel is that when people are there, we feel very vulnerable because people do not comply with everything we say, we feel that we are all more exposed to possible contamination” (N7). These testimonies show that the nurses’ perception of the visitors is that they are unaware of and are not motivated to comply with the measures to prevent transmission of infection. Thus, it will be important to understand which strategies can be implemented to overcome this. However, this assessment may be influenced by nurses’ fear of the contamination and dissemination of COVID-19 and, once again, by the structural conditions that were not the most appropriate for the safety and control of infection.

Some participants also reported that this absence of visits simplifies/facilitates the provision and care (3.5): “For us nurses, I think it’s more positive in this sense, it facilitates our provision of care. We are not often interrupted by questions from the family, which is normal, and I would do the same in their place. Currently… we don’t have to do this management on the family” (N3). This is something “favorable, as sometimes we have families that also get in the way a little. In addition, it ends, they often ask many questions and have some discussion with us. It improved slightly, and it was favorable” (N9). These responses reveal that some professionals are not aware of the practice of family-centered nursing care and have a view of the family as an element external to the care practice, a view that has long been abandoned, which is evidenced by the emergence of several models of care delivery and recommendations at this level [10]. Traditionally, family-centered care is more common in the pediatric area, yet there are already initiatives in adult care with positive results [35]. There is an urgent need for a paradigm shift with regard to the participation of the family in care, so structural and procedural conditions should be rethought to obtain better health outcomes.

In this same sense, the category *Absence of family-centered care* (3.6) emerged, in which nurses revealed that the absence of family has the implication “…not to work with the family… hospitalization can be an opportunity to, in addition to resolving an acute situation, working in an acute hospital can also be an opportunity to verify many situations in which we nurses can, if we want, have this role, and not all of us consider the length of stay as an opportunity to leverage change in some aspects of the health of patients and their families” (N1). Within this category were two subcategories: *Noninvolvement of the family in care* (3.6.1): “…regarding the provision of care, the issue of partnership was jeopardized by restricting visits; the impossibility of involving the family in the monitoring of the patient,” (N5); and difficulty in training the caregiver (3.6.2) to home care: “Additionally, as a negative, the fact that we cannot… train the caregiver to take care of, for example, feeding the patient by tube, to manage the therapeutic and drug regimen, is also unfavorable” (N5). Considering that the evidence has shown the advantages and positive results of the implementation of family-centered care, namely, in the safety, quality, and satisfaction with the health care provided, this implication of the absence of families can bring negative consequences for patients, families, and health care providers [16,24,30,31,32].

The category *Communication with the affected family* (3.7) was mentioned by the nurses interviewed: “This communication also becomes complicated because we can’t evaluate well who is on the other side, what information and how we should give it” (N4). This implication results in the lack of information about patients and difficulty in planning their discharge, two subcategories we identified. The *Lack of information about patients* (3.7.1) was referred to as “Negative also in the anamnesis, in the collection of data both for us and for the medical side, it became very complicated to obtain information. We have to make phone calls more often because there was no other way to contact the family, so there was a lot of information that failed to reach us. We didn’t have access to the patient’s history in terms of daily life behavior or clinical history, which is a negative aspect of the absence of families” (N9). In this sense, a study on the application of open visits in two large elderly wards showed that this change in the visit policy improved communication and trust between families and health professionals [32]. Thus, it is important to note that communication failures in health care have been widely recognized as a cause of health errors [36]. Nevertheless, the errors associated with the lack of communication with the family require more research. In the second subcategory, *Difficulty in planning discharge* (3.7.2): “the difficulty is also related to discharge planning. The presence of the family before was facilitating, and its absence with this restriction made it difficult” N8. On this subject, a systematic review of randomized trials that compared an individualized discharge plan with a nonindividualized discharge routine revealed that a structured discharge plan that is individualized to each patient shortens the initial hospitalization time and decreases readmissions in the elderly, and it also contributes to increased patient satisfaction with the health care received [37]. Another study reported that the period after hospital discharge is a vulnerable time for patients; approximately half of adults suffer a medical error after hospital discharge, and 19–23% suffer an adverse event, most often related to medication [29].

It was also mentioned that the telephone inquiries by the relatives led to the nurses having *Less time to provide care* (3.8): “Although it is positive on the one hand, as we get some quietness within our routines, we end up losing that time because we have to answer the phone and we end up wasting time talking to family members” (N8). Effectively, as previously mentioned, time management, given the ratios of nurses in Portugal, is problematic and is reflected in these circumstances.

The interviewees also reported that the restrictive measures on visits “was negative, led to impersonality of care…” (N6), leading us to identify the category *Less person-centered care* (3.9). It is widely recognized that person-centered care has the potential to generate significant benefits for the health and health care of all people, including better access to care, better health and clinical outcomes, better health literacy and self-care, greater satisfaction with care, greater job satisfaction, greater efficiency of services, and reduction in overall costs [38].

Considering the number of categories on each side, the implications for care practice were more negative than positive. On the positive side are categories related to the nurses’ fear of infection and the alleged nonadherence of families to infection prevention measures. Regarding the categories identified as negative for care practice, many are related to the safety of care and the patient, such as communication with the affected family (subcategories: lack of information about patients and difficulty planning discharge), and others were related to the quality of care, and all have the potential to have negative consequences for the patient and their families in the medium and long term.

Regarding the categories related to care practice, what stands out above all is a dichotomy between interviewees, some considering the absence of the family as something positive, others considering it something negative, and some were even ambivalent. Regarding these aspects, nurses with a lower degree of schooling were the ones who identified more positive aspects of care practice from the separation of families from hospitalized patients. This finding reveals that training could make an important contribution to the practice of person- and family-centered care.

## 5. Conclusions

Although the family has an important role in contributing to patient safety, it can be seen from the interviewees’ discourses that the provision of nursing care is still focused on the patient, based on the biomedical model, and the family is not seen as a target of care. With the COVID-19 pandemic and the implementation of hospital visit restrictions attempting to mitigate its spread, the investment in care delivery methods in partnership with patients and families was even more jeopardized. In this study, it was seen that there had been mostly negative consequences for patients, relatives, and the care process itself.

The nurses interviewed were unanimous in stating that for the hospitalized patient, the implications of the visit restriction policy due to COVID-19 were negative in several domains, including mental health, resistance to adherence to the therapeutic regimen, and the feeling of insecurity, which call into question the quality and safety of care.

Similarly, there was a consensus in stating that these restrictive measures were negative for the families, with an impact on their well-being, increasing their anxiety and their appeals for information since they could not obtain it easily. This makes family members feel insecure about the care provided to their hospitalized relatives. These findings also indicate that the quality and safety of care have been jeopardized, as has the application of family-centered care.

In turn, regarding the care practice, there was ambiguity in the responses of the participating nurses, as they reported positive and negative implications. The positive ones refer to the fact that they have more time to provide care to the patient due to the absence of families because this absence simplifies the provision of care and because they consider that families do not follow the rules for infection prevention. The negative implications identified are consistent with all the implications for patients and relatives mentioned above and have a strong potential to lower the safety and quality of care. In this sense, it was found that in addition to dichotomous evaluations among nurses about the presence of the family in care practice, there were nurses who gave ambivalent answers on the subject. The nurses with less schooling were the ones who seemed more positive about the absence of families, and nurse training may make an important contribution to the practice of person- and family-centered care. It is urgent to change the paradigm in hospital care, obtaining conditions for the participation of the family in the care and obtaining better health outcomes. For this purpose, in addition to training professionals, structural and procedural conditions must be rethought and reorganized.

Considering the number of categories on each side, the implications for patients, relatives, and care practice are far more negative than positive and have a strong potential to cause safety events in the short to long term, also jeopardizing the quality of care. Objectives pursued and some goals achieved in these matters are at risk and suffered setbacks during the pandemic period, not only in the context of safety and quality of care but also regarding person- and family-centered care. Health decision-makers, health professionals in general, and nurses, in particular, should know and consider these implications of restrictive measures of visits and their consequences that may last beyond the hospitalization period or even the pandemic period and should seek strategies to mitigate these implications.

The limitations of this study are the fact that it was conducted during the pandemic period, which may have conditioned the responses due to the fear of spreading the virus that causes COVID-19.

More studies are needed, including replication of this study in the postpandemic period, studies in other areas of expertise, and quantitative studies on the medium- and long-term implications of these measures, their impact on the health of individuals and families, and strategies to mitigate them.

## Figures and Tables

**Figure 1 ijerph-19-08991-f001:**
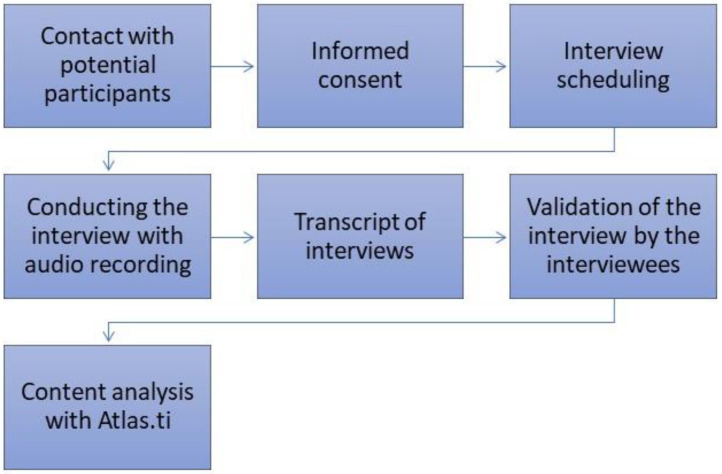
Method flow.

**Figure 2 ijerph-19-08991-f002:**
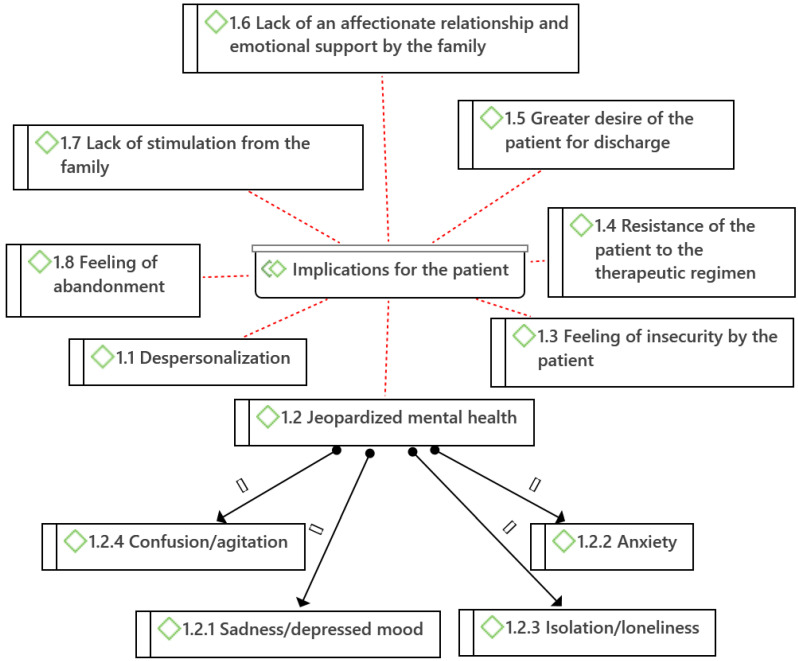
Categories and subcategories of implications for the patient of hospital visit restriction measures (Atlas.ti^®^).

**Figure 3 ijerph-19-08991-f003:**
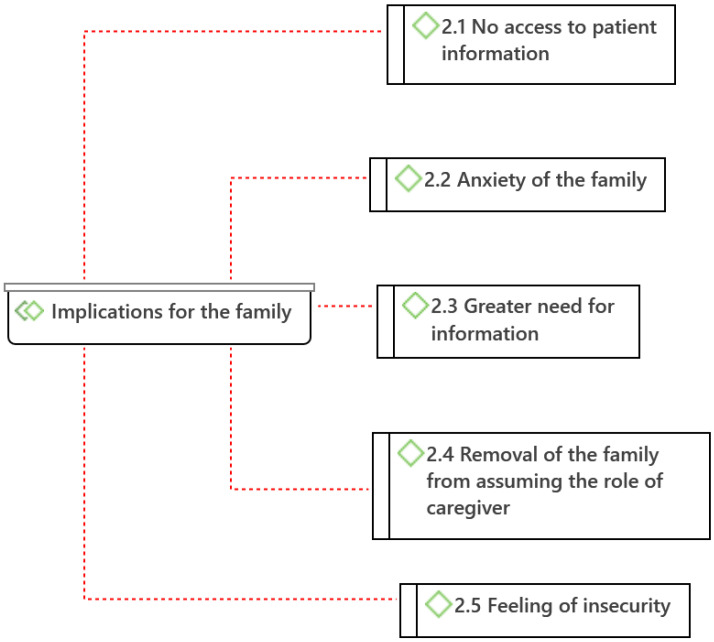
Categories of implications for the family by hospital visit restriction measures. (Atlas.ti^®^).

**Figure 4 ijerph-19-08991-f004:**
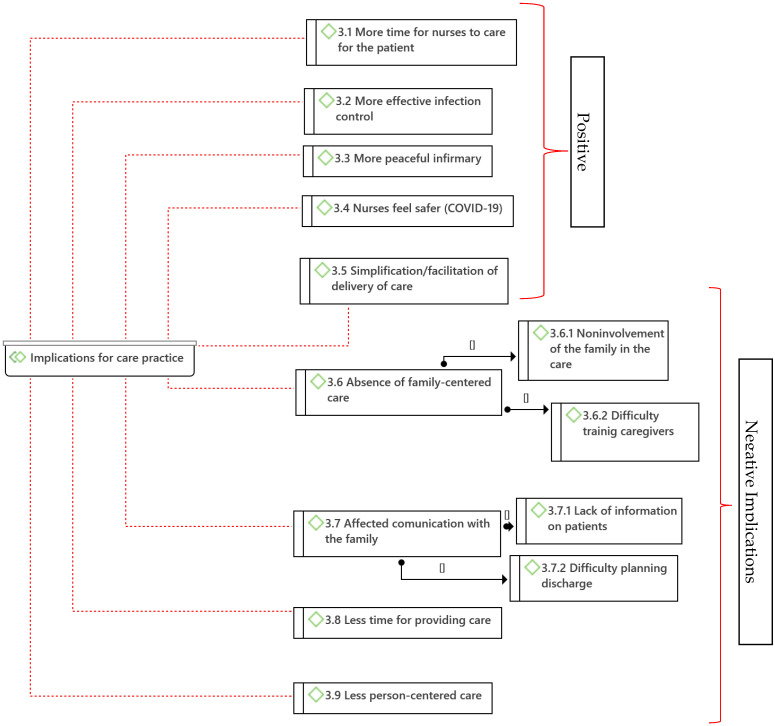
Categories of implications for care practice caused by hospital visit restriction measures (Atlas.ti^®®®^).

## Data Availability

Not applicable.

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
