# Peer review of "The Implications of Family Members’ Absence from Hospital Visits during the COVID-19 Pandemic: Nurses’ Perceptions"

_ijerph, 2022, doi:10.3390/ijerph19158991_

Round 1

Reviewer 1 Report

The manuscript entitled, The implications of family members’ absence from hospital visits during the COVID-19 pandemic: Nurses’ perception needs more polishing before it can be accepted for publication.

Main Comments:

1. The sample size of nurses should be increased. Ten nurses do not represent the perception of all nurses. As you increase the sample size, the nurses' perceptions should differ also.

2. The manuscript should be written scientifically. There are so many ellipses in the discussion. The parts of the manuscript must be coherent.

Minor comments:

3. Lines 93 to 97: Materials and Methods should be written in sentences than bullets.

4. Line 100: The range of age of nurses' experience is broad. It wouldn’t validate the results. Break it down into decades.

5. Make all the figures stand alone, organized, and cleaner. Spellcheck.

6. Line 140: Enumerate the categories and reflect them in the figure.

7. Make the discussion more scientific. The sentences should be written scientifically.

Author Response

Dear reviewer!

First of all, we want to thank you for your comments in order to improve our article. We have tried to respond to all suggestions and we believe that the article has significantly improved with the suggested changes.

We go on to respond to each suggestion:

  1. The sample size of nurses should be increased. Ten nurses do not represent the perception of all nurses. As you increase the sample size, the nurses' perceptions should differ also.

After the seventh interview, no new data were found, that is, new categories, so after ten interviews we consider that data saturation has been reached. (line 113-114)

  1. The manuscript should be written scientifically. There are so many ellipses in the discussion. The parts of the manuscript must be coherent.

We try to improve the discussion according to your suggestions. 

Minor comments:

  1. Lines 93 to 97: Materials and Methods should be written in sentences than bullets.
  2.  

Thus, the inclusion criteria werebeing a practicing nurse during the COVID-19 pandemic period;practicing in inpatient internal medicine and adult surgical services in one of the four hospitals in northern Portugal (in any specialty);having at least 4 years’ experience in these hospitals (this criterion arises from the authors' understanding as the minimum experience to be able to contribute to the study considering that it is a topic that may not be the focus of attention of less experienced professionals.);and being available to participate. (line 102-111)

  1. Line 100: The range of age of nurses' experience is broad. It wouldn’t validate the results. Break it down into decades.

One participant under the age of 30, six between the ages of 30 and 40, two participants between the ages of 40 and 50 and one participant over sixty. (line 116-118)

  1. Make all the figures stand alone, organized, and cleaner. Spellcheck.

All figures have been changed as suggested.

  1. Line 140: Enumerate the categories and reflect them in the figure.

All figures have been changed as suggested. Category numbering has been included in the texto as well.

  1. Make the discussion more scientific. The sentences should be written scientifically.

We try to improve the discussion according to your suggestions

The article was further revised by a certified translator.

We believe and hope to have improved the article according to your expectations and suggestions, we believe so.

Best regards

Reviewer 2 Report

This study is about the implication of restricting visits to hospitalized individuals. 

[Introduction] 

1. The criteria for applying restriction visits to hospitalized individuals may vary depend on the status of the patients. Could you explain why this study is important? 

2. Could you add research questions in lines 82 through 83?

[Materials and method]

3. Additional description of the criteria for the inclusion criteria is needed. Why is the number of years of work for nurses four years? Does the workplace include special department such as organ transplant wards or infectious medicine wards? 

4. Additional description of how the data saturation was reached is needed. 

5. Could you explain the process of reaching the data saturation with 10 interviewees (interview questions, number of interviews, etc.)  

Author Response

Dear reviewer!

First of all, we want to thank you for your comments in order to improve our article. We have tried to respond to all suggestions and we believe that the article has significantly improved with the suggested changes.

We go on to respond to each suggestion:

[Introduction] 

  1. The criteria for applying restriction visits to hospitalized individuals may vary depend on the status of the patients. Could you explain why this study is important? 

The criteria for applying restrictions on visits due to COVID-19 varied from institution to institution. There were also times when there were statutory enactments of this restriction that was blind and did not consider the consequences of these measures. This study may help to ensure that the measures to restrict visits in response to this or any other situation are appropriate and in the best interest of the patient and the quality and safety of care. (line 83-88)

  1. Could you add research questions in lines 82 through 83?

In this way, the study aims to answer the following research questions:

What are the implications of the absence of family members in hospital visits during the COVID-19 pandemic identified by nurses?

How do nurses assess these implications for the patient, the family and care practice? (line 88-92)

[Materials and method]

  1. 1 Additional description of the criteria for the inclusion criteria is needed. Why is the number of years of work for nurses four years?

This criterion arises from the authors' understanding as the minimum experience to be able to contribute to the study considering that it is a topic that may not be the focus of attention of less experienced professionals. (line 107-110)

  • Does the workplace include special department such as organ transplant wards or infectious medicine wards? 

Yes (line 106)

  1. Additional description of how the data saturation was reached is needed. 

After the seventh interview, no new data were found, that is, new categories, so after ten interviews we consider that data saturation has been reached. (line 113-114)

  1. Could you explain the process of reaching the data saturation with 10 interviewees (interview questions, number of interviews, etc.)  

Data were collected through individual semistructured interviews between June and September 2020 to learn the perceptions of nurses about the implications of the absence of families from hospitals during the COVID-19 pandemic with the following questions:

What do you see as positive and negative in restricting visits?

What was favorable and unfavorable of the restriction of visits for nurses?

What was favorable and unfavorable of restricting visits for patients? (line 120-125)

We believe and hope to have improved the article according to your expectations and suggestions, we believe so.

Best regards

Round 2

Reviewer 1 Report

The manuscript should be improved. 

Lines 89-91: What are these? Better explain.

Lines 95 to98: I think this is an Experimental Design, and should be in the results and discussion part.

For materials and methods, please put Experimental or Method flow and improve your writings here. This part is very crude.

Lines 194 to 195, 249: There are still ellipses (…). This is not a scientific statement. It should be rephrased and summarized in such a way that it will become a comprehensive statement for the readers. Please check other parts, too.

Please put Ethical Guidelines or the participants agree to some extent that they will be part of this research.

Author Response

Dear reviewer!

First of all, we want to thank you for your comments in order to improve our article. We have tried to respond to all suggestions and we believe that the article has significantly improved with the suggested changes.

We go on to respond to each suggestion:

For materials and methods, please put Experimental or Method flow and improve your writings here. This part is very crude.

  • We introduced a method flow (line 135) and improvements to the methodology were made.

Please put Ethical Guidelines or the participants agree to some extent that they will be part of this research.

  • “Ethical and legal principles were respected. The study was authorized by the Joint Ethics Committee of the Hospital and University Center of Porto and the Biomedical Sciences Institute Abel Salazar (Instituto Ciências Biomédicas Abel Salazar - ICBAS) of the University of Porto (Universidade do Porto - UP). All procedures performed with the participants respected anonymity, confidentiality, and informed consent, as well as the Declaration of Human Rights of Helsinki.” (Line 139-144)

Lines 89-91: What are these? Better explain.

  • These are research questions introduced at the suggestion of the other reviewer.

Lines 95 to98: I think this is an Experimental Design, and should be in the results and discussion part.

  • This is a qualitative study that aims to know the perception of nurses about the reality under study through semi-structured interviews. We do not apply any intervention.

Lines 194 to 195, 249: There are still ellipses (…). This is not a scientific statement. It should be rephrased and summarized in such a way that it will become a comprehensive statement for the readers. Please check other parts, too.

  • We have changed statements that can be considered ellipses.

The article was further revised by a certified translator.

We believe and hope to have improved the article according to your expectations and suggestions, we believe so.

Best regards
